# Investigation of the Relationship between Sensory-Processing Skills and Motor Functions in Children with Cerebral Palsy

**DOI:** 10.3390/children10111723

**Published:** 2023-10-24

**Authors:** Serhat Erkek, Çiğdem Çekmece

**Affiliations:** 1Department of Occupational Therapy, Yalova State Hospital, Baglarbası, Yalova 77100, Turkey; serokerkek@hotmail.com; 2Section of Occupational Therapy, Department of Therapy and Rehabilitation, Vocational School of Kocaeli Health Services, Kocaeli University, Umuttepe Campus, Izmit 41380, Turkey

**Keywords:** cerebral palsy, sensory processing, occupational therapy

## Abstract

The main purpose of this study is to examine the relationship between sensory-processing skills and gross motor functions, bimanual motor functions, and balance in children with cerebral palsy (CP). A total of 47 patients between the ages of 3 and 10, diagnosed with CP, who received or applied for treatment in our physical therapy and rehabilitation unit were included in the study. Sensory profiling (SP), assisting hand assessment (AHA), the Gross Motor Function Measure-66 (GMFM-66), and the Pediatric Berg Balance Scale (PBBS) were used in the evaluation of the children with CP who participated in the study. The Gross Motor Function Classification System (GMFCS) was used to classify the children based on functional abilities and limitations, and the Manual Ability Classification System (MACS) was used to classify the children based on manual dexterity. The SP parameters were compared with AHA, GMFM-66, and PBBS results, and with GMFCS and MACS levels. Statistically significant relationships were found between AHA and SP, PBBS, and SP and between GMFM-66 and SP (*p* < 0.05). Our study shows that there are some disorders in sensory processing in children with CP. We think that sensory evaluations should be included in the CP rehabilitation program.

## 1. Introduction

Research focusing on the functional limitations and the challenges experienced by children with cerebral palsy (CP) has indicated that factors such as muscle strength, trunk control, balance, and postural stability play significant roles in influencing these children’s ability to perform daily activities [1,2,3]. Furthermore, the limited number of existing studies in the literature suggest that functional deficiencies observed in children with CP could potentially be linked to issues with multisensory integration and sensory-processing deficits [4,5].

Sensory processing involves the neural system’s processing of sensory data that originates from both the external environment and the body’s internal mechanisms. Sensory processing encompasses the functions of sensory receptors, as well as the functions of the peripheral and central nervous systems. The brain takes on the role of organizing, integrating, synthesizing, and utilizing this information to comprehend experiences and to orchestrate appropriate responses. This information-processing mechanism enables individuals to react to sensory stimuli automatically, effectively, and comfortably [6]. Sensory processing forms the basis for learning, perception, and action. Due to individual variations, there may be differences in the senses, including the tactile, auditory, visual, taste, and smell senses. In addition to the differences in these senses, there may be differences in some sensory processes, such as those of the proprioceptive and vestibular systems. These sensory differences can negatively affect development and functional abilities in behavioral, emotional, motor, and cognitive areas [7].

Sensory-processing disorders are characterized by the impairment of one or more sensory systems, leading to maladaptive behavior and motor responses [8,9]. The literature extensively outlines functional issues that are associated with sensory-processing disorders. In 2001, Parham and Mailloux [10] delineated five functional challenges that are linked to sensory-processing disorders: reduced proficiency in social skills and participation in play activities; decreased frequency, duration, or complexity of adaptive responses; impaired self-esteem and/or self-efficacy; insufficient proficiency in adaptive or daily living skills; and difficulties in the development of fine, gross, and sensorimotor skills.

Postural problems or problems with voluntary movements (dyspraxia) may be seen in children with motor disorders that originate from sensory-processing disorders. With postural problems, a child may have difficulty stabilizing his or her body during movement or in adapting to environmental requirements during rest. Observations include difficulties in maintaining the necessary muscle balance between flexion and extension for a specific activity, inappropriate muscle tone, and issues with midline crossing. Activities that require fine motor skills may be accompanied not only by poor motor-planning skills but also by postural disorders that result from weak muscle tone in the shoulders and the upper body [11,12]. The literature indicates that approximately 90% of children with CP have sensory-processing problems, such as tactile and proprioceptive disorders [13].

Depending on their sensory-processing skills, the participation of children with CP in the activities of daily living (ADL) is affected at different levels. It has been stated that sensory-processing disorders can make life difficult for children with CP by affecting their functionality in care activities that require bilateral upper-extremity use, such as eating, playing, dressing, and showering [14]. The literature on this topic indicates a connection between sensory-processing challenges in children with CP and their abilities in sensory integration and ADL, including tasks such as playing, eating, sleeping, dressing, and engaging in leisure and school-related activities [14]. A study conducted by Jeanette Curry stated that preschool children with CP may be at high risk for somatosensory disorders that can significantly affect hand functions and pointed out the presence of sensory-processing deficits in children with CP. In addition, it has been stated that the occurrence of different problems together restricts a child’s participation in activities, thus affecting the child’s sensory experience and causing secondary sensory disorders [15].

Although the participation of children with CP in ADL is especially related to movement and posture disorders, accompanying sensory problems, loss of cognitive skills, perception difficulties, and behavioral disorders negatively affect the developmental process and the children’s functional independence [16]. Different studies on CP have shown that difficulties with tactile sense, proprioception, vestibular sense, and vision are the most common sensory-processing problems [17,18]. Research indicates that a majority of individuals with CP encounter sensory challenges [19] together with motor impairments, encompassing tactile [20], proprioceptive [21], and visual [22] deficits. Nevertheless, despite a growing emphasis in the scientific literature over the past two decades, the assessment of sensory issues in children with CP is not widely integrated into rehabilitation practices. This omission can potentially result in misunderstandings regarding the underlying factors that contribute to challenges in motor-performance domains [23]. Unfortunately, although these sensory-based disorders are not commonly considered as the primary feature of CP, they are frequently seen in this population. They remain an area that has not been adequately addressed [18,24,25]. However, given the effects of these frequently seen sensory disorders on motor performance and participation in activities, it is important to evaluate sensory processing correctly.

Although sensory-processing disorders are frequently observed in children with CP, this problem can often be overlooked during treatment planning. This situation also limits the use of sensory-based interventions based on sensory processing in CP rehabilitation. The principal objective of this study is to ascertain the sensory profiles of children with CP, based on their motor levels. Additionally, it seeks to explore the correlation between sensory-processing skills and the gross motor functions, bimanual motor functions, and balance reactions of children with CP.

## 2. Materials and Methods

The study included patients aged between 3 and 10 years who were being treated for CP in the Department of Physiotherapy and Rehabilitation at the University of Kocaeli or who were diagnosed with CP, had no additional disease that could interfere with the evaluation, and had not undergone surgery in the last 6 months. Children with severe mental retardation, muscle contracture or bone deformity, autism spectrum disorder, or communication problems, and children diagnosed with hyperactivity and attention deficit disorder were excluded from the evaluation. The parents/caregivers of all subjects of the study were informed about the study and provided informed, written consent before the subjects’ participation in the study. This study was approved by the Kocaeli University Ethical Committee (KÜ GOKAEK-2022/19.11).

The sociodemographic information about the patients with CP who participated in the study was recorded. In the form, the children’s gender, age, and CP type were specified. We utilized the Gross Motor Function Classification System (GMFCS) to categorize the children according to their functional abilities and limitations. Additionally, we employed the Manual Ability Classification System (MACS) to classify them based on their capacities to independently grasp objects and their requirements for assistance or adaptations in performing daily manual activities. In our study, we employed sensory profiling (SP) to assess sensory-processing skills; Assisting Hand Assessment (AHA) to evaluate bimanual motor functions; and the Gross Motor Function Measure-66 (GMFM-66) to the assess gross motor functions of all participating patients. Additionally, balance was evaluated using the Pediatric Berg Balance Scale (PBBS). SP scores were compared with AHA, GMFM-66, and PBBS evaluation results.

The evaluations were carried out at a quiet, normal room temperature in a bright and ventilated environment where the children could feel comfortable and safe. Materials such as balls, mats, parallel bars, steps, and thresholds—required for GMFM-66 and PBBS evaluations—were used. For AHA evaluation, steps were provided to support the children’s feet and a table and chair suitable for their heights were provided. The certified therapist who made the assessments was allowed to sit directly opposite the children. With the toys included in the AHA kit and suitable for the children’s age groups, the environment was prepared to allow the children to perform all of their upper-extremity functions bimanually. A 15-min filming was carried out for every/each children, with the camera placed behind the therapist at an appropriate angle. Then, the video images were watched and scoring was completed and recorded.

### 2.1. Assessment Scales

#### 2.1.1. Sensory Profiling (SP)

Sensory profiling was designed by an occupational therapist. The questionnaire assessing the sensory processing of children aged 3–10 was filled out by their parents or caregivers. It consisted of 125 items. The survey consisted of 14 subsections under 3 main headings. These main topics were sensory processing, modulation, and emotional–behavioral responses. Sensory processing consisted of 6 sections, modulation consisted of 5 sections, and behavioral–emotional responses consisted of 5 subsections. The first part determined the level of problems in sensory processing. The second part specified the status of endurance and tone-related traits, movements, and activities for modulation. The last section evaluated behavioral and emotional responses to sensory inputs. The questionnaire was scored as 1, 2, 3, 4, and 5, respectively, according to the following answers—always, often, sometimes, rarely, never—that were given for each item. In our study, the parameters of the questionnaire—vestibular processing, tactile processing, sensory processing related to endurance and tone, regulations related to movement and body position, sensory registration, sensory seeking, sensory sensitivity, and sensory avoidance were evaluated and correlations were made according to these parameters [26].

##### Vestibular Processing

Vestibular processing refers to the functions associated with the processing of the vestibular sense. The vestibular system plays a crucial role in our perception of movement in space, awareness of body position, maintenance of postural tone and balance, as well as in coordinating our eye movements during spatial motion. Within the inner ear, the vestibular component comprises the semicircular canals, the utriculus, and the sacculus. The semicircular canals primarily regulate dynamic balance, while the utriculus is responsible for maintaining static balance in response to gravity.

##### Tactile Processing

The tactile sense involves receiving information from the skin in response to environmental stimuli. It plays a critical role in the development of skills such as grasping, body awareness, social interaction, motor planning, and fine motor skills. A deficiency in tactile processing can lead to sensory insecurity and motor challenges in children.

##### Sensory Processing Related to Endurance and Tone

Sensory processing related to endurance and tone encompasses elements associated with the regulation of proprioception. The proprioceptive sense involves feedback regarding the coordination of muscles, joints, and the brain’s understanding of movement in relation to time and space. Proprioception is crucial for the development of posture, motor planning, and body awareness. Challenges in processing proprioceptive input can result in slow movement, clumsiness, and increased energy expenditure in children.

##### Regulations Related to Movement and Body Position

Regulations related to movement and body position typically refer to a set of rules or guidelines that dictate how children should move their bodies or position themselves in a particular context or setting. They contain regulations related to the modulation of vestibular sense and proprioceptive sense.

##### Sensory Registration

Sensory registration is the mechanism through which children react to and focus on sensory stimuli in their surroundings. Initially, the nervous system detects sensory input, and once this information is stored in memory, the nervous system ascribes significance to the new data by comparing it with previously encountered stimuli or visual cues. In a low sensory register, children appear uninterested or unaware of the sensory stimuli around them and do not respond.

##### Sensory Seeking

Children with sensory seeking are motivated to experience an unusual amount or type of sensation. They participate tirelessly in activities to provide their bodies with stimulation from many sensory modalities. Certain children who seek sensory experiences might encounter difficulties in executing smooth and coordinated movements. Sensory-seeking children may exhibit signs of discomfort within their own bodies. They have poor posture and a lack of stability.

##### Sensory Sensitivity

Sensory sensitivity pertains to the level of awareness that children possess with respect to each of their sensory modalities, encompassing sight, sound, taste, smell, touch, and pain. Every individual exhibits a unique spectrum of sensitivity, and their outward expressions and responses to these sensitivities vary accordingly. With sensory sensitivity, children respond more quickly, intensely, and for longer durations to sensory stimuli. Sensory sensitivity can occur in a single sensory system (e.g., tactile defensiveness or gravitational insecurity) or in multiple sensory systems.

##### Sensory Avoidance

Sensory avoidance is the avoidance of stimuli by people who have sensory sensitivity and are aware of this situation. People with sensory avoidance try to find a solution by showing that they are disturbed by the stimulus. Sensory avoidance is a behavior or a coping mechanism that is observed in individuals with sensory-processing difficulties, particularly those people who experience sensory sensitivity or sensory overload. In sensory avoidance, individuals actively try to minimize their exposure to certain sensory stimuli that they find overwhelming or distressing. This could involve avoiding places, situations, or activities that trigger sensory discomfort. For example, someone with sensory avoidance may avoid noisy environments or situations with strong odors to reduce sensory overload. Sensory avoidance is a way for individuals to manage their sensory challenges and find comfort in their sensory environment.

#### 2.1.2. Assisting Hand Assessment (AHA)

AHA measures how effectively individuals use both hands together to perform bimanual tasks. Bimanual performance is of paramount importance, as daily activities often require the use of both hands. AHAs are observational and sensitive to change. The bimanual activities for which the test is scored are semi-structured to allow interaction with the relevant therapist, depending on the subject’s age. Results from AHAs are used to guide interventions and measure changes over time. The test consists of two stages. First, the person making the assessment (e.g., a therapist or a physician) sits directly opposite the child, and an environment is prepared that will allow the child to perform all upper-extremity functions bimanually with toys suitable for the child’s age group from the AHA kit. Approximately 15 min of filming is carried out, with the camera previously placed behind the therapist at an appropriate angle. Following the filming, the therapist watches the recording and scores each activity on a scale of 1–4 (4: effective use; 3: partially effective use; 2: ineffective use; and 1: inability to use) [27].

#### 2.1.3. Gross Motor Function Measure-66 (GMFM-66)

GMFM-66 is an observational clinical scale that is sensitive to impairments. It evaluates gross motor functions in children with CP. The scale assesses motor function in five dimensions related to developmental gross motor function milestones [28].

#### 2.1.4. Pediatric Berg Balance Scale (PBBS)

The PBBS is the version of the Berg Balance Test used in adults that is adapted for children. It is a scale that functionally evaluates balance, and it consists of 14 questions with parameters such as sitting with support, sitting without support, transferring, standing without support, and turning [29].

#### 2.1.5. Gross Motor Function Classification System (GMFCS)

The GMFCS is a widely used evidence-based classification system. It was developed as a simple method to classify children with CP, based on their functional skills and limitations. It is based on performing activities and movements independently, focusing on sitting, transferring, and mobility. The GMFCS includes five levels and four age groups, with differences based on functional limitations, the need for a hand-held assistive walking device (e.g., a walker, crutches, or a cane) or a wheeled mobility device, and, to a lesser extent, the quality of movement [30].

#### 2.1.6. Manual Ability Classification System (MACS)

The MACS test assesses how children with CP utilize their hands during daily activities involving object manipulation. The MACS is categorized into five levels, each of which is based on children’s capacity to independently grasp objects and their requirement for assistance or adaptations in carrying out daily manual activities. Additionally, the MACS identifies distinctions between two consecutive levels to precisely determine the appropriate level that corresponds to children’s abilities [31].

### 2.2. Statistical Analysis

The data obtained from the research was analyzed using the IBM Statistical Package for Social Science Statistics (SPSS) 25.0 statistical package program. In analyzing the data, as descriptive statistical methods, the number of units (n) and the percentage rate (%), the arithmetic mean (X), and standard deviation (SD) were used in the quantitative parametric variables. In the quantitative non-parametric variables: the median (Xort), the lower value, and the upper value were used. In the comparison of categorical/qualitative variables: Pearson’s chi-square (chi-square or x^2^) or Fisher’s exact tests were used, depending on the suitability of the data. The suitability of the quantitative variables to normal distribution was examined with the Shapiro–Wilk test. The parametric test was used for the quantitative variables that were found to be normally distributed and the non-parametric test was used for the quantitative variables that were found to be non-normally distributed. The independent-samples *t*-test was applied to the parametric quantitative variables, and the Mann–Whitney U test was applied to the non-parametric quantitative variables. In the correlation analyses performed to examine the relationship between the variables, all statistical results were evaluated at the 95% confidence interval, and the significance was evaluated at the *p* < 0.05 level.

## 3. Results

Our study considered 55 children with CP who were between the ages of 3 and 10. Eight of these 55 children were excluded from the study because they did not meet the inclusion criteria (one patient had a surgical history, three patients did not fit the age range, two patients had mental retardation, and two patients had severe contracture). Of the 47 children with CP who were included in the study, 28 (59.6%) were boys and 19 (40.4%) were girls; 53.2% were between the ages of 3–7 and 46.8% were between the ages of 8–10. All 47 children with CP had spastic-type CP (22 were diplegic, 9 were hemiplegic, and 16 were quadriplegic). According to the GMFCS levels, it was determined that one child was level I, 20 children were level II, four children were level III, 12 children were level IV, and 10 children were level V. Based on the EBSS levels of the children, seven children were at level I, 18 children were at level II, nine children were at level III, two children were at level IV, and 11 children were at level V. The patients’ demographic information is provided in Table 1.

The average values of the parameters of vestibular processing, tactile processing, sensory processing related to endurance and tone, regulations related to movement and body position, sensory registration, sensory seeking, sensory sensitivity, and sensory avoidance for the children with CP who participated in the study are provided in Table 2.

Table 3 shows the distribution of SP parameters according to CP type. It was determined that children with diplegic- and hemiplegic-type CP performed better than children with quadriplegic-type CP in the parameters of sensory processing related to endurance and tone and sensory registration. In the sensory sensitivity parameter, children with hemiplegic-type CP performed better than children with quadriplegic-type CP.

Table 4 shows the distribution of sensory problems according to CP type.

### 3.1. Findings Regarding SP Parameters

When Table 5 is examined, it can be observed that the CP children showed a definite difference according to SP in the parameters of vestibular processing in 76.6% of the cases, sensory processing related to endurance and tone in 83.0% of the cases, and regulation related to movement and body position in 85.1% of the cases. Furthermore, it was determined that 84.8% of the children with CP fell within the definite difference range in terms of sensory registration, 34.0% fell within the definite difference range in terms of sensory seeking, 40.4% fell within the definite difference range in terms of sensory sensitivity, and 34.0% fell within the definite difference range in terms of sensory avoidance scores, according to the SP parameters.

### 3.2. Results of Correlation Analysis of Parameters

The correlation analysis of the evaluation parameters of the children with CP included in the study is provided in Table 6.

Based on the results of the correlation analysis, a strong negative correlation (*p* < 0.001) was found between the GMFCS level and the sensory registration and sensory processing related to endurance and tone parameters, and a weak negative correlation (*p* < 0.05) was found between the GMFCS level and the sensory sensitivity parameter (Table 6).

A moderate negative relationship (*p* < 0.001) was detected between the MACS level and the sensory registration and sensory processing related to endurance and tone parameters. A statistically significant relationship was found between the MACS level and the sensory sensitivity parameter at a weak level (*p* < 0.05) in the negative direction (Table 6). There was a strong positive (*p* < 0.001) relationship between the GMFM-66 level and sensory registration and sensory processing related to endurance and ton” and a moderately positive (*p* < 0.001) positive relationship between the GMFM-66 level andthe sensory sensitivity parameter. A weak, positive, statistically significant relationship was determined between the GMFM-66 level and the regulations related to movement and body position parameter (*p* < 0.05) (Table 6). A strong positive correlation was found between AHA and sensory prelated to endurance and tone and sensory registration (*p* < 0.001), as well as a moderate positive correlation between AHA and sensory sensitivity (*p* < 0.001). However, a weak positive relationship was identified between the regulation related to movement and body position parameter and AHA (*p* < 0.05) (Table 6).

A strong positive correlation (*p* < 0.001) was observed between the PBBS value and the parameters sensory registration and sensory processing related to endurance and tone. Additionally, a weak positive statistically significant relationship (*p* < 0.05) was found between the PBBS value and the parameters vestibular processing and regulation related to movement and body position (Table 6).

## 4. Discussion

Children with CP experience functional difficulties not only due to muscle tone and weak postural control but also because of sensory issues [13]. Although sensory-processing disorders are commonly observed in children with CP, they often go unnoticed [32]. Sensory-processing difficulties can impact a child’s daily activities, emotional well-being, and motor functions [33].

Children with CP struggle to explore their surroundings and engage in activities involving different movements. Motor difficulties, including uncontrolled movement patterns and insufficient postural control, influence both the amount and effectiveness of proprioceptive and vestibular sensory input. These motor challenges can, consequently, influence a child’s body awareness, postural control, motor planning, bilateral coordination, and cognitive development, potentially leading to secondary impairments and atypical sensory responses. It is noted that vestibular and proprioceptive inputs, which feed into motor responses, can lead to specific challenges in children with different types of CP. In hemiplegic-type CP children, these challenges are particularly associated with motor planning and bilateral coordination difficulties. In quadriplegic-type CP children, sensory-perception or sensory-modulation issues may arise due to these inputs. In contrast, diplegic-type CP children can experience gravity insecurity that is linked to vestibular stimulation [34].

Studies have shown that approximately 90% of children with CP exhibit sensory disorders [32]. Sensory-modulation difficulties, sensory-discrimination difficulties, and weak sensory-registration skills in children with CP lead to challenges in arousal levels, affecting motor functions, attention, motivation, planning, and behavioral organization [35,36]. In the literature, it has been shown that children with CP experience problems in at least one of the SP parameters (sensory registration, sensory sensitivity, sensory seeking, and/or sensory avoidance) [37]. Accurately identifying the various sensory-processing issues observed in children with CP will contribute to the rehabilitation process by complementing conventional treatments and interventions.

While deficiencies in the sensory systems of children with CP have been widely reported in the literature [31,32,38,39], we noticed that only a few studies mentioned the presence of sensory-processing disturbances in children with CP [24,32,40]. There needs to be more research in the literature regarding the relationship between sensory-processing skills in children with CP and their gross motor functions, bimanual performance, and balance. This study examined the relationship between sensory-processing skills in children with CP and their gross motor functions, bimanual performance, and balance. Our study provides a valuable contribution by offering a comprehensive assessment in this regard.

Our study evaluated the sensory-processing skills of children with CP using SP. Consistent with the literature, it was determined that all of the studied patients exhibited sensory disorders in all of the SP parameters. These sensory-processing disorders are suggested to be associated with an abnormal mechanism within the sensory–motor network in children with CP, potentially as a result of decreased thalamocortical projections [24]. It is also suggested that these structural deficiencies may jeopardize tactile and somatosensory information-processing in children with CP [41]. A study conducted by Pavão and Rocha of 43 CP children, using SP, revealed disturbances in parameters that were similar to those observed in our study [36]. Another study supported this finding, indicating that children with CP exhibit significant impairments in sensory processing, which define their sensory profiles. During physical activities, such children also display low resistance and muscle weakness [42].

In our study, the relationship between different types of SP and sensory processing was examined. When comparing the types of CP, a statistically significant relationship was observed for the parameters of sensory processing related to endurance and tone, sensory sensitivity, and sensory registration. In our assessments, it was determined that children with diplegic and hemiplegic types of CP exhibit better indicators in these parameters and possess better processing skills than children with quadriplegic-type CP. One plausible reason for the comparatively lower occurrence of these characteristics in the diplegic and hemiplegic forms of CP, as opposed to other types, could be associated with the mobility patterns exhibited by children with these specific CP types. Children with diplegic and hemiplegic CP often engage in activities such as crawling, maneuvering among furniture, or using assistive devices. These modes of movement offer proprioceptive feedback through the use of the upper extremities. This can support the development of endurance, tone, movement, body position, and sensory-modulation skills. In the literature, it is also observed that, in addition to the sensory-registration parameter, there are favorable results for children with diplegic CP in terms of sensory sensitivity [43].

In our analysis of the relationship between gross motor function and sensory-processing skills, a significant relationship was found between the levels of GMFM-66 and GMFCS and the parameters of sensory processing related to endurance and tone, sensory registration. and sensory sensitivity. A significant relationship was identified between GMFM-66 levels and the regulation related to movement and body position parameter: As GMFM-66 scores decrease and GMFCS levels increase, impairments in these parameters were more commonly observed. While there have been studies examining the relationship between sensory-processing skills and GMFCS levels in CP patients in the literature [44,45], no study was found that investigated the relationship with the GMFM-66 scale.

In the current literature, it is mentioned that approximately 40% to 70% of children with SP have sensory-discrimination disorders [46,47,48]. Impairments in perception and registration related to tactile and proprioceptive sensory input can affect manipulation skills in children with CP [49]. Wingert et al. [48] associated this issue with impairments observed in the SP’s parietal and frontal cortical somatosensory regions, which affect fine motor coordination and tactile shape, and sharp discrimination deficits that influence sensory sensitivity. While there have been studies demonstrating sensory problems in children with SP and their impact on ADLs, no studies have examined the relationship between upper-extremity bimanual use and sensory processing. Our study assessed the bimanual functions of children with SP using AHA. AHA is the only assessment that measures how effectively a paralyzed hand is used for bimanual activities, which could be a crucial aspect of hand function for these children and which has been validated for use in children with CP [27]. In our study, a significant relationship was found between AHA and the parameters of sensory processing related to endurance and tone, sensory registration, sensory Sensitivity, and regulation related to movement and body position. Therefore, there are significant differences between the levels of bilateral coordination and fine motor skills and the levels of sensory processing in children with CP. In a study using the short sensory profile and the MACS in the literature, a correlation similar to the result of our study was demonstrated between the total sensory-processing score and the MACS levels [35]. Our study suggests that when evaluating upper-extremity functions in children with CP, a consideration of sensory-processing skills is also necessary, as motor performance in bimanual activities is affected, based on the sensory parameters associated with AHA.

It has been stated that vestibular and proprioceptive processing deficiencies could contribute to poor muscle control. They may also contribute to balance issues, potentially exacerbating the problem in children with CP [43,50]. This is attributed to the decreased thalamocortical projections from the thalamus to S1, which create deficits in somatosensory processing in individuals with CP [24]. No literature was found comparing sensory-processing skills with PBBS scores. However, in some of the studies examined, it was stated that PBBS scores increased in children with CP whose vestibular-processing skills increased due to the sensory-integration therapies applied. [51,52]. In our analysis of the relationship between PBBS scores, which functionally evaluate balance, and sensory-processing skills, a significant relationship was found for the parameters of vestibular processing, sensory processing related to endurance and tone, regulations related to movement and body position, sensory registration, and sensory sensitivity. This indicates that as PBDT scores decrease, children with SP exhibit weakened vestibular-processing skills and postural-control responses, decreased activity levels, increased sensitivity to different types of sensory stimuli, and diminished sensory-perception skills.

While sensory-related impairments such as compromised tactile, proprioceptive, kinesthetic, and pain perception are not regarded as primary characteristics of CP, they frequently manifest in this group and often receive insufficient attention. Evaluating sensory processes in children with CP and including restricted areas in rehabilitation studies is important in terms of contributing to motor performance and functionality. In our study, it was determined that sensory-processing skills in children with CP are associated with motor function, balance, and upper-extremity bimanual use. According to the results obtained from the study, sensory-processing disorders in children with CP vary, but problems related to vestibular sensory processing and proprioception are commonly experienced. Sensory-modulation disorders are more commonly observed in children with CP, particularly in the parameters of sensory registration and sensory sensitivity. Sensory registration can affect perception skills in children with CP, while sensory sensitivity can impact their levels of participation in activities. We believe that raising awareness among rehabilitation teams and families about sensory disorders can lead to a more effective process in SP rehabilitation, and assessing sensory modulation can positively impact the treatment process.

## 5. Conclusions

Considering observations made in this study, the sensory-processing difficulties evident in children with CP underscore the need for implementing therapies that incorporate a sensory-oriented approach, together with motor stimulation. Such interventions aim to enhance functional improvements in the daily lives of such children.

## Figures and Tables

**Table 1 children-10-01723-t001:** Demographic information of the patients.

		Boy		Girl		Total	
		N	%	N	%	N	%
Age	3–10 years	28	59.6	19	40.4	47	100
CP Type	Diplegic	11	39.3	11	57.9	22	46.8
	Hemiplegic	7	25.0	2	10.5	9	19.1
	Quadriplegic	10	35.7	6	31.6	16	34.0
GMFCS	I	1	3.6	0	0	1	2.1
	II	15	53.6	5	26.3	20	42.6
	III	2	7.1	2	10.5	4	8.5
	IV	4	14.3	8	42.1	12	25.5
	V	6	21.4	4	21.1	10	21.3
MACS	I	5	17.9	2	10.5	7	14.9
	II	10	35.7	8	42.1	18	38.3
	III	5	17.9	4	21.1	9	19.1
	IV	2	7.1	0	0	2	4.3
	V	6	21.4	5	26.3	11	23.4

**Table 2 children-10-01723-t002:** The average values of the parameters of SP.

	Vestibular Processing	Tactile Processing	Sensory Processing Related to Endurance and Tone	Regulations Related to Movement and Body Position	Sensory Registration	Sensory Seeking	Sensory Sensitivity	Sensory Avoidance
Total (N = 47) $\bar{x}$ ± SD/Xmed (Lower Value–Upper Value)	41.0 ± 4.50	77 (27–89)	24.4 ± 9.73	30.8 ± 5.79	44 (25–77)	96.0 ± 14.65	75.2 ± 10.11	116 (79–142)

Those parameters with normal distribution are shown as $\bar{x}$ ± SD, and those with non-normal distribution are shown as Xmed (lower value–upper value). $\bar{x}$: arithmetic mean; SD: standard deviation; Xmed: median.

**Table 3 children-10-01723-t003:** Distribution of SP parameters according to CP type.

	CP Type				
	DP Median (IQR)	HP Median (IQR)	QP Median (IQR)	*p* ^a^	Post-Hoc Test ^b^
Vestibular Processing	42 (38.8–46)	40 (37.5–44)	39 (37.3–43)	0.530	-
Tactile Processing	78.5 (69.8–85)	78 (66–85)	75.5 (70–81.3)	0.714	-
Sensory Processing Related to Endurance and Tone	27.5 (20.5–36.3)	30 (20–36)	18 (13.8–19.8)	**0.001**	1, 2 > 3
Regulations Related to Movement and Body Position	31.5 (30–35.5)	31 (28.5–35)	28.5 (23.5–33.5)	0.082	-
Sensory Registration	52.5 (42.5–61.3)	53 (41.5–59)	36 (31–40.8)	**<0.001**	1, 2 > 3
Sensory Seeking	96 (86.5–107.3)	94 (82–106)	96.5 (91.5–103.8)	0.901	-
Sensory Sensitivity	78 (68.8–85)	80 (75–87)	70 (63.5–75.8)	**0.023**	2 > 3
Sensory Avoidance	118.5 (96–124.3)	127 (103–129.5)	111.5 (99–121.3)	0.398	-

DP: diplegic, HP: hemiplegic, QP: quadriplegic; IQR: interquartile range; ^a^ Kruskal–Wallis test; ^b^ Dunn’s test.

**Table 4 children-10-01723-t004:** Distribution of sensory problems according to CP type.

		Diplegic CP		Hemiplegic CP		Quadriplegic CP	
		N	%	N	%	N	%
Vestibular Processing	Definite Difference	15	68.2	8	88.9	13	81.2
	Possible Difference	5	22.7	1	11.1	2	12.5
	Typical Performance	2	9.1	-	-	1	6.3
Tactile Processing	Definite Difference	1	4.5	2	22.2	2	12.5
	Possible Difference	6	27.3	3	33.3	3	18.8
	Typical Performance	15	68.2	4	44.4	11	68.7
Sensory Processing Related to Endurance and Tone	Definite Difference	16	72.7	7	77.8	16	100.0
	Possible Difference	2	9.1	-	-	-	-
	Typical Performance	4	18.2	2	22.2	-	-
Regulations Related to Movement and Body Position	Definite Difference	17	77.3	8	88.9	15	93.8
	Possible Difference	2	9.1	1	11.1	1	6.3
	Typical Performance	3	13.6	-	-	-	-
Sensory Registration	Definite Difference	16	76.2	7	77.8	16	100.0
	Possible Difference	2	9.5	2	22.2	-	-
	Typical Performance	3	14.3	-	-	-	-
Sensory Seeking	Definite Difference	8	36.4	4	44.4	4	25
	Possible Difference	6	27.2	1	11.1	7	43.2
	Typical Performance	8	36.4	4	44.4	5	31.3
Sensory Sensitivity	Definite Difference	8	36.4	1	11.1	10	62.4
	Possible Difference	6	27.2	5	55.6	5	31.3
	Typical Performance	8	36.4	3	33.3	1	6.3
Sensory Avoidance	Definite Difference	9	40.9	2	22.2	5	31.3
	Possible Difference	2	9.1	1	11.1	4	25.0
	Typical Performance	11	50.0	6	66.7	7	43.8
Total		22	46.9	9	19.1	16	34.0

**Table 5 children-10-01723-t005:** Distribution of SP parameters by gender.

		Boy		Girl		Total	
		N	%	N	%	N	%
Vestibular Processing	Definite Difference	19	67.9	17	89.5	36	76.6
	Possible Difference	6	21.4	2	10.5	8	17.0
	Typical Performance	3	10.7	0	0	3	6.4
Tactile Processing	Definite Difference	3	10.7	2	10.5	5	10.6
	Possible Difference	6	21.4	6	31.6	12	25.5
	Typical Performance	19	67.9	11	57.9	30	63.9
Sensory Processing Related to Endurance and Tone	Definite Difference	22	78.6	17	89.5	39	83.0
	Possible Difference	2	7.1	0	0	2	4.3
	Typical Performance	4	14.3	2	10.5	6	12.8
Regulations Related to Movement and Body Position	Definite Difference	24	85.7	16	84.2	40	85.1
	Possible Difference	3	10.7	1	5.3	4	8.5
	Typical Performance	1	3.6	2	10.5	3	6.4
Sensory Registration	Definite Difference	22	81.5	17	89.5	39	84.8
	Possible Difference	3	11.1	1	5.3	4	8.7
	Typical Performance	2	7.4	1	5.3	3	6.5
Sensory Seeking	Definite Difference	11	39.3	5	26.3	16	34.0
	Possible Difference	5	17.9	9	47.4	14	29.8
	Typical Performance	12	42.9	5	26.3	17	36.2
Sensory Sensitivity	Definite Difference	11	39.3	8	42.1	19	40.4
	Possible Difference	9	32.1	7	36.8	16	34.0
	Typical Performance	8	28.6	4	21.1	12	25.5
Sensory Avoidance	Definite Difference	10	35.7	6	31.6	16	34.0
	Possible Difference	3	10.7	4	21.1	7	14.9
	Typical Performance	15	53.6	9	47.3	24	51.1

**Table 6 children-10-01723-t006:** Correlation analysis of evaluation parameters.

		AHA *	PBBS *	GMFS-66 **	GMFCS *	MACS *
PBBS	r	0.739	-	0.873	−0.933	−0.777
	*p*	**<0.001**	-	**<0.001**	**<0.001**	**<0.001**
GMFS-66	r	0.816	0.873	-	−0.901	−0.841
	*p*	**<0.001**	**<0.001**	-	**<0.001**	**<0.001**
GMFCS	r	−0.738	−0.933	−0.901	-	0.767
	*p*	**<0.001**	**<0.001**	**<0.001**	-	**<0.001**
MACS	r	−0.891	−0.777	−0.841	0.767	-
	*p*	**<0.001**	**<0.001**	**<0.001**	**<0.001**	-
Vestibular Processing	r	0.270	0.356	0.270	−0.238	−0.270
	*p*	0.067	**0.014**	0.066	0.107	0.066
Tactile Processing	r	0.168	0.285	0.263	−0.177	−0.218
	*p*	0.260	0.052	0.074	0.235	0.142
Sensory Processing Related to Endurance and Tone	r	0.611	0.705	0.738	−0.723	−0.63
	*p*	**<0.001**	**<0.001**	**<0.001**	**<0.001**	**<0.001**
Regulations Related to Movement and Body Position	r	0.293	0.323	0.367	−0.269	−0.280
	*p*	**0.046**	**0.027**	**0.011**	0.067	0.056
Sensory Registration	r	0.678	0.745	0.771	−0.71	−0.678
	*p*	**<0.001**	**<0.001**	**<0.001**	**<0.001**	**<0.001**
Sensory Seeking	r	−0.114	0.007	−0.028	0.057	0.075
	*p*	0.445	0.964	0.851	0.703	0.616
Sensory Sensitivity	r	0.453	0.513	0.513	−0.449	−0.404
	*p*	**<0.001**	**<0.001**	**<0.001**	**0.002**	**0.005**
Sensory Avoidance	r	0.224	0.232	0.266	−0.154	−0.209
	*p*	0.129	0.116	0.071	0.301	0.158

*: Spearman’s Rho correlation coefficient. **: Pearson’s correlation coefficient.

## Data Availability

The data presented in this study are available on request from the corresponding author. The data are not publicly available due to privacy and ethical restrictions.

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
