# Peer review of "Investigation of the Relationship between Sensory-Processing Skills and Motor Functions in Children with Cerebral Palsy"

_children, 2023, doi:10.3390/children10111723_

Round 1

Reviewer 1 Report

A valuable work. Its design is appropriate and its language is understandable enough.

A reference regarding the validity and reliability of the tests used in your country should be added.

In Table 3, it is not appropriate to abbreviate the CP types as 1,2,3. DP, HP, QP can be made or written openly.

In Table 4, the number 1 has shifted in the first row. It is doubtful whether Table 4 actually needs to exist. General distribution information should either be included in Table 1 or deleted, it is not the primary purpose of the study.

The distribution of parameters according to gender is given in Table 6, but no analysis or comment has been made regarding whether there is a significant difference between the percentages between girls and boys.

Reviewer 2 Report

Great manuscript.

However, because I am not very familiar with SP parameters. Please consider providing brief definitions for each of the eight SP parameters.

Page 12 first complete paragraph. There is one sentence that needs to be translated to English.

no concerns

Reviewer 3 Report

The present study is of particular interest in the field of rehabilitation of pediatric cerebral palsy. The impairment of sensory processing is a very important and little explored aspect in the literature. In this scenario, this study is considered important and overall well organized and described.

Unfortunately, I noted a point that needs to be clarified by the Authors not only in the review, but especially in the text:

1. In the Study design section, the Authors report the following: "Sensory Profile (SP)

Designed by occupational therapist Dunn. The questionnaire assessing the sensory

processing of children aged 3-10 is filled out by caregivers. It consists of 125 items. The

survey consists of 14 subsections under 3 main headings".

Validated and standardized tools already exist for the study of the sensory profile (e.g. SPM-P, SPM, Sensory Profile 2). Why did the authors decide to create a new questionnaire ad hoc? What does the questionnaire created by the occupational therapist add to those already available in the literature? Why weren't standardized tools used?

This is a very important point which, as I have already said, will also have to be clarified in the text.

For the rest the study is certainly well done.

Round 2

Reviewer 1 Report

It is enough for Table 4 to be included in the text, it would be better if you delete the table. Other corrections are sufficient. Thanks.
